EMBO
Molecular Medicine

# Supraphysiological levels of GDF11 induce striated muscle atrophy

David W Hammers[1,2] (ID), Melissa Merscham-Banda[1,2], Jennifer Ying Hsiao[3], Stefan Engst[3], James J Hartman[3] & H Lee Sweeney[1,2,*] (ID)

## Abstract

Growth and differentiation factor (GDF) 11 is a member of the transforming growth factor β superfamily recently identified as a potential therapeutic for age-related cardiac and skeletal muscle decrements, despite high homology to myostatin (Mstn), a potent negative regulator of muscle mass. Though several reports have refuted these data, the *in vivo* effects of GDF11 on skeletal muscle mass have not been addressed. Using *in vitro* myoblast culture assays, we first demonstrate that GDF11 and Mstn have similar activities/potencies on activating p-SMAD2/3 and induce comparable levels of differentiated myotube atrophy. We further demonstrate that adeno-associated virus-mediated systemic overexpression of GDF11 in C57BL/6 mice results in substantial atrophy of skeletal and cardiac muscle, inducing a cachexic phenotype not seen in mice expressing similar levels of Mstn. Greater cardiac expression of *Tgfbr1* may explain this GDF11-specific cardiac phenotype. These data indicate that bioactive GDF11 at supraphysiological levels cause wasting of both skeletal and cardiac muscle. Rather than a therapeutic agent, GDF11 should be viewed as a potential deleterious biomarker in muscle wasting diseases.

**Keywords** activin receptor; cachexia; cardiac atrophy; muscle mass; myostatin
**Subject Categories** Ageing; Musculoskeletal System

## Introduction

The transforming growth factor (TGF)β superfamily of signaling proteins consists of a number of ligands having diverse effects. In mammals, there are more than 30 superfamily members that are currently classified into 4 subfamilies based on sequence similarities (Mueller & Nickel, 2012): TGFβs, the bone morphogenic proteins (BMPs)/growth and differentiation factors (GDFs), the activins/inhibins/nodals, and "others", all having various roles in development, tissue maintenance, tissue repair, and disease. The GDF member, myostatin (Mstn; also known as GDF8), is of particular importance/interest in skeletal muscle as a potent negative regulator of muscle mass (McPherron *et al*, 1997), and strategies to inhibit its activity have received much focus for the treatment of degenerative and wasting diseases of skeletal muscle (Sumner *et al*, 2009; Morine *et al*, 2010a; Zhou *et al*, 2010). GDF11, which shares 90% homology with Mstn in its mature signaling peptide, has been the subject of recent reports proposing anti-aging effects in heart (Loffredo *et al*, 2013), skeletal muscle (Sinha *et al*, 2014), and brain (Katsimpardi *et al*, 2014). These data have generated controversy about the physiological effects and therapeutic potential of GDF11, as highlighted in a recent pair of contending viewpoint articles (Harper *et al*, 2016; Walker *et al*, 2016).

Like many other members of the TGFβ superfamily, Mstn and GDF11 are both translated as a precursor protein with a secretion signal sequence, a propeptide domain, and a mature peptide domain, with a disulfide bond between two mature peptide domains (forming a dimer). The proteins are further processed by cleavage of the mature peptides from their respective propeptides, though they remain bound in a latent complex such that the propeptides prevent the mature dimer from becoming an active ligand. This latent complex is secreted and can be activated by proteolysis of the propeptides by BMP-1/tolloid metalloproteinase (Wolfman *et al*, 2003; Ge *et al*, 2005), which releases the mature dimer to become a free ligand.

Cell signaling induced by TGFβ superfamily ligands is mediated though several type I and type II receptors that can dimerize in different combinations, based on affinities to the ligand and receptor expression by the cell (Mueller & Nickel, 2012). Mstn and GDF11 are ligands for the activin type II B receptor (ActRIIB; a TGFβ type II receptor). Their binding to ActRIIB recruits either activin-like kinase (ALK) 4 or ALK5 type I receptors to the complex, activating the canonical ActRIIB signaling pathway through phosphorylation of SMAD2/3, dimerization with SMAD4, and activation of transcriptional activity (Derynck & Zhang, 2003). The TGFβ superfamily can also activate non-canonical, less elucidated pathways that involve mitogen-activated protein kinase (MAPKs), including p38 MAPK

1  Department of Pharmacology & Therapeutics, University of Florida College of Medicine, Gainesville, FL, USA
2  Myology Institute, University of Florida College of Medicine, Gainesville, FL, USA
3  Cytokinetics Inc., San Francisco, CA, USA
   *Corresponding author. Tel: +1 352 273 9416; Fax: +1 352 392 3558; E-mail: lsweeney@ufl.edu

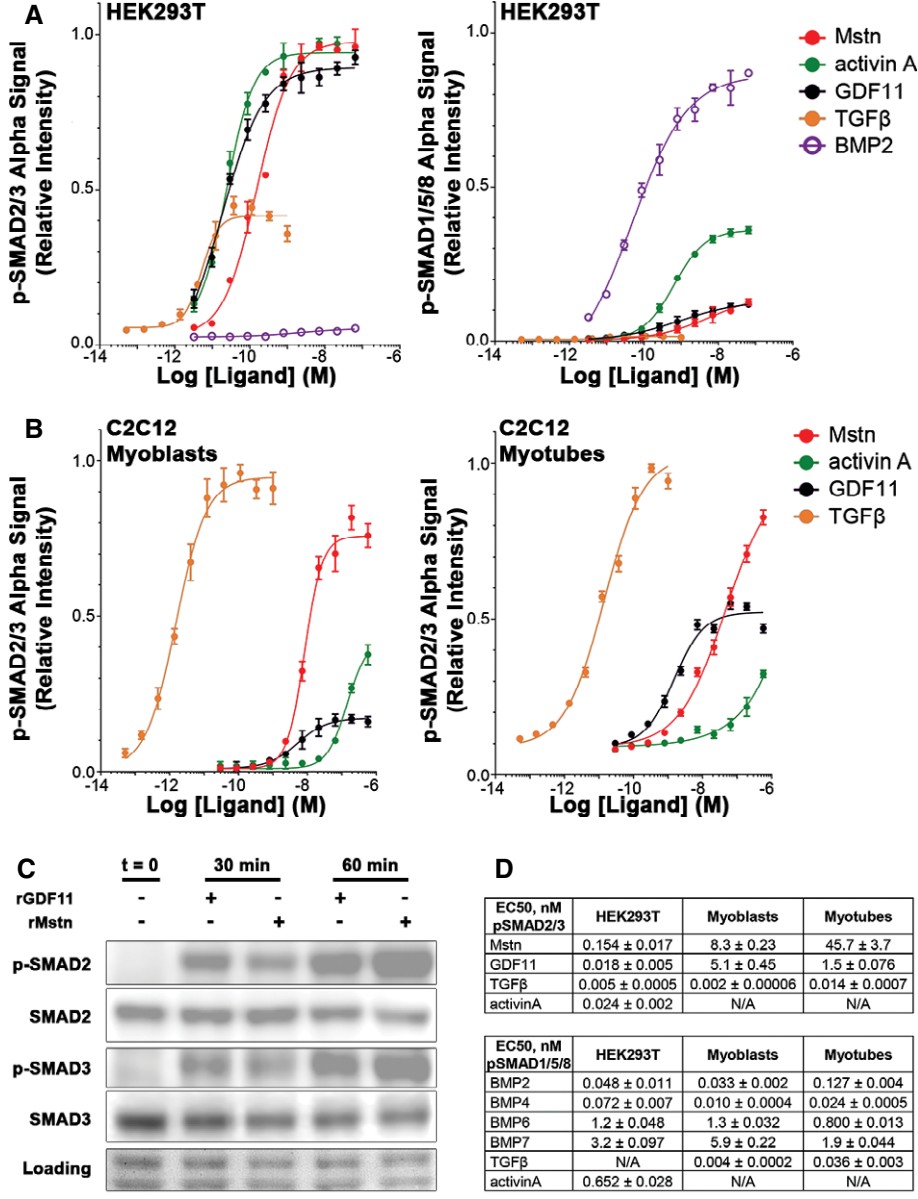

**Figure 1.  Activation of SMAD pathways by TGFβ superfamily ligands *in vitro*.**

A   Relative p-SMAD2/3 (left) and p-SMAD1/5/8 (right) response, evaluated by AlphaLISA signal, of HEK293T cells after 1-h exposure to recombinant myostatin (Mstn), activin A, GDF11, TGFβ, and BMP2.

B   Relative p-SMAD2/3 response of C2C12 myoblasts (left) and myotubes (right) after 1-h exposure to Mstn, activin A, GDF11, and TGFβ.

C   Phosphorylation of SMAD2 and SMAD3 in differentiated C2C12 myotubes following stimulation with recombinant Mstn or GDF11 for 30 and 60 min, as detected by immunoblotting. Equal loading is verified by Ponceau Red staining.

D   EC$_{50}$ values (in nM) for p-SMAD2/3 and p-SMAD1/5/8 responses of HEK293T, C2C12 myoblasts, and C2C12 myotubes to the above listed ligands, as well as p-SMAD1/5/8 response to BMP4, BMP6, and BMP7.

Data information: Values are displayed as mean ± SEM; *n* = 4 for all data points.

and extracellular regulated kinase (ERK) pathways (Derynck & Zhang, 2003).

While it is well known that Mstn, through activation of the canonical SMAD2/3 pathway, negatively affects skeletal muscle mass *in vitro* and *in vivo* (McPherron *et al*, 1997; Trendelenburg *et al*, 2009), the effects of GDF11 on muscle have been less clear, despite the highly similar structure and signaling activation. Unlike

the highly muscular Mstn knockout mouse (McPherron *et al*, 1997), ablation of *GDF11* is perinatal lethal and results in skeletal patterning defects (McPherron *et al*, 1999). Additionally, disruption of *GDF11* in skeletal muscle results in no change in phenotype (McPherron *et al*, 2009), suggesting muscle-derived GDF11 has a minor, if any, physiological role. This ambiguity of GDF11 function in striated muscle has led to controversy over whether recombinant

GDF11 therapy can reverse age-related cardiac hypertrophy (Loffredo *et al*, 2013) or skeletal muscle regenerative defects (Sinha *et al*, 2014). Indeed, several reports have emerged in response to these data, indicating non-specific GDF11 detection methods (Egerman *et al*, 2015; Rodgers & Eldridge, 2015; Smith *et al*, 2015), irreproducibility of GDF11 decline with age (Egerman *et al*, 2015; Rodgers & Eldridge, 2015), prevention of cardiac hypertrophy (Smith *et al*, 2015), and the enhancement of muscle regeneration (Egerman *et al*, 2015; Hinken *et al*, 2016).

From this controversy, it is clear that the biological effects of GDF11 are not yet clearly delineated. The aim of the present report is to characterize the effects of bioactive GDF11 on differentiated striated muscle tissue. We demonstrate that recombinant GDF11 is a more potent activator of SMAD2/3 than Mstn and induces atrophy in differentiated myotubes. In addition, we report that systemic overexpression of full-length GDF11 in C57BL/6 mice promotes a severe wasting phenotype in both skeletal and cardiac muscle. While Mstn and GDF11 similarly affect skeletal muscle mass *in vivo*, GDF11 overexpression leads to lethality.

## Results

### Recombinant GDF11 induces SMAD2/3 activation and atrophy in C2C12 myotubes

To assess relative potencies of TGFβ superfamily ligands on SMAD activation *in vitro*, we evaluated canonical signaling for several ligands, including Mstn and GDF11. In HEK293T cells, phosphorylation of SMAD2/3 (p-SMAD2/3) is potently stimulated by Mstn, GDF11, activin A, and TGFβ (Fig 1A left), while phosphorylation of SMAD1/5/8 (p-SMAD1/5/8) shows relatively modest stimulation by only activin A (Fig 1A right), especially in comparison with known stimulators of this pathway, such as BMP2 (a BMP type II receptor ligand). We also verified the involvement of the ActRIIB in the responses of Mstn, GDF11, and activin A using a murinized version of the BYM338 antibody that targets ActRIIB (Fig EV1A and B). In the presence of antibody, stimulation of HEK293T cells with GDF11, Mstn, or activin A resulted in > 100-fold reduced potencies for all three ligands. By comparison, the potency and amplitude of TGFβ stimulation remained unchanged, thus verifying the requirement of ActRIIB for the potencies of affected ligands.

In differentiated C2C12 myotubes (Fig 1B right), TGFβ is a potent stimulator of p-SMAD2/3, while Mstn and GDF11 demonstrate less potency ($\sim 10^3$ less potent than TGFβ), with GDF11 being more potent than Mstn ($\sim$30-fold), and activin A showing even less potency. The maximum level of p-SMAD2/3, however, is much reduced for GDF11 compared to Mstn. C2C12 myoblasts demonstrate a similar response to these ligands (Fig 1B left), though the relative amplitude of p-SMAD2/3 in response to GDF11 stimulation over differentiated cells is significantly decreased, while the potency is similar to that observed in myotubes. Effective induction of p-SMAD2 and p-SMAD3 in myotubes after 30 and 60 min of exposure to Mstn or GDF11 was also demonstrated by immunoblotting (Fig 1C). Phosphorylation of SMAD1/5/8 in C2C12 myoblasts or myotubes is not affected by GDF11 or Mstn, though BMP ligands potently stimulate this pathway (Fig EV1C). The measureable $EC_{50}$s of ligands in these experiments are shown in Fig 1D.

As it has been previously demonstrated that recombinant GDF11 can inhibit myoblast differentiation *in vitro* (Trendelenburg *et al*, 2009; Egerman *et al*, 2015), we sought to determine whether sustained exposure of GDF11 induces atrophy in differentiated myotubes. Thus, at seven days following induction of differentiation of C2C12 myoblasts, we added media supplemented with no ligand, of recombinant (r)Mstn, rGDF11, or rTGFβ (50 ng/ml; depicted in Fig 2A), and analyzed myotube diameter after 3 days of treatment to ensure steady-state effects of ligand exposure are observed. GDF11-treated myotube diameter (−40%) was nearly identical in size as Mstn (−36%) and TGFβ-treated myotubes (−34%), compared to control myotubes (Fig 2B–D). The observed myotube atrophy induced by Mstn and GDF11 does not appear to be due to loss of myonuclei, as myotube nuclear content (normalized to length of myotubes) is actually increased by treatment (Fig 2E). Phosphorylation of SMAD3 via Mstn and GDF11 can also be found in these myotubes (Fig 2F), albeit to a lower degree than the acute stimulation shown in Fig 1C, as is typical for cellular responses to chronic stimuli. Therefore, we conclude that exposure of GDF11 does negatively affect myotube size *in vitro* in response to the potent activation of p-SMAD2/3.

### Systemic GDF11 overexpression induces skeletal and cardiac muscle atrophy *in vivo*

The effects of supraphysiological GDF11 were tested *in vivo* (scheme depicted in Fig 3A) by treating 12-week-old C57BL/6 male mice with full-length murine GDF11 expressed in the liver using the liver-specific α1-anti-trypsin promoter (with ApoE enhancer) packaged into AAV2/8 (referred hereafter as AAV8.GDF11). Robust expression of the transgene was evident within days, as AAV8.GDF11-treated mice required euthanasia 7 days following treatment after losing over 35% of their body mass (Fig 3E). The liver exhibited clear expression of full-length and monomeric GDF11 [Fig 3B; analyzed using R&D Systems' clone 743833 mouse mAb (Egerman *et al*, 2015; Smith *et al*, 2015)]. Serum levels of ~60 μg/ml were estimated from the monomeric band [equating to ~3.3 mg/kg for a 23 g mouse (using the standard mouse blood volume of 0.055 ml/g bwt); Fig 3C], while in the quadriceps, monomeric GDF11 content was in the order of 0.001% of the total muscle protein (Fig 3D). Interestingly, each AAV8.GDF11 sample exhibits a ~25 kDa immunoreactive band (denoted with * in Figs 3 and EV2) that is particularly prominent in the serum (immunoreactive with anti-mouse IgG, yet cannot be depleted by a combination of protein A/G and protein L-coated agarose beads; Fig EV2D). Also included in these experiments was a group of mice treated with an identical AAV2/8 dose of a construct containing full-length murine Mstn (AAV8.Mstn). However, due to substantial disparity in expression (detected using anti-Mstn C-terminal REGN459 mouse mAb; Fig EV2) between these two viral treatments, AAV8.Mstn data from this seven-day treatment cohort are only included in the Expanded View data.

The severe wasting phenotype caused by this robust systemic increase in GDF11 includes skeletal muscle mass loss of 21% in the soleus, 30% in the extensor digitorum longus, 23% in the tibialis anterior (TA), 23% in the gastrocnemius, and 26% in the quadriceps, as well as a 29% loss of heart mass (Fig 3F). This atrophy is evident at the myocyte level, as TA fiber and cardiomyocyte size is

**Figure 2. GDF11 induces myotube atrophy *in vitro*.**

A, B   C2C12 myoblasts were differentiated for seven days, then control (*n* = 5), myostatin (Mstn; *n* = 4), growth differentiation factor 11 (GDF11; *n* = 5), or transforming growth factor β (TGFβ; *n* = 3) supplemented media (50 ng/ml) was added for 3 days (depicted in A) before PFA fixation and staining with myosin heavy chain (MHC) to evaluate myotube diameter (B). The scale bars represent 50 μm.

C, D   Myotube diameter data (*n* = 200–400 myotubes from 3 to 5 different trials) are depicted as a histogram of diameter distribution (C) and as mean diameter (mean ± SEM; D) by treatment.

E   Myotube nuclear content was quantified as DAPI-positive nuclei per μm of myotube length for control, Mstn, and GDF11 treatment groups.

F   Phosphorylation of SMAD3 in control, Mstn, and GDF11 treatment groups, as detected by immunoblotting. Equal loading is verified by Ponceau Red staining.

Data information: Data are depicted as histogram of distribution (C) or mean ± SEM (D, E). Statistical analysis was performed using one-way ANOVA analysis with Tukey's HSD *post hoc* test (non-connecting letters indicate *P* < 0.05 between groups) and effect size presented as eta-squared ($\eta^2$).

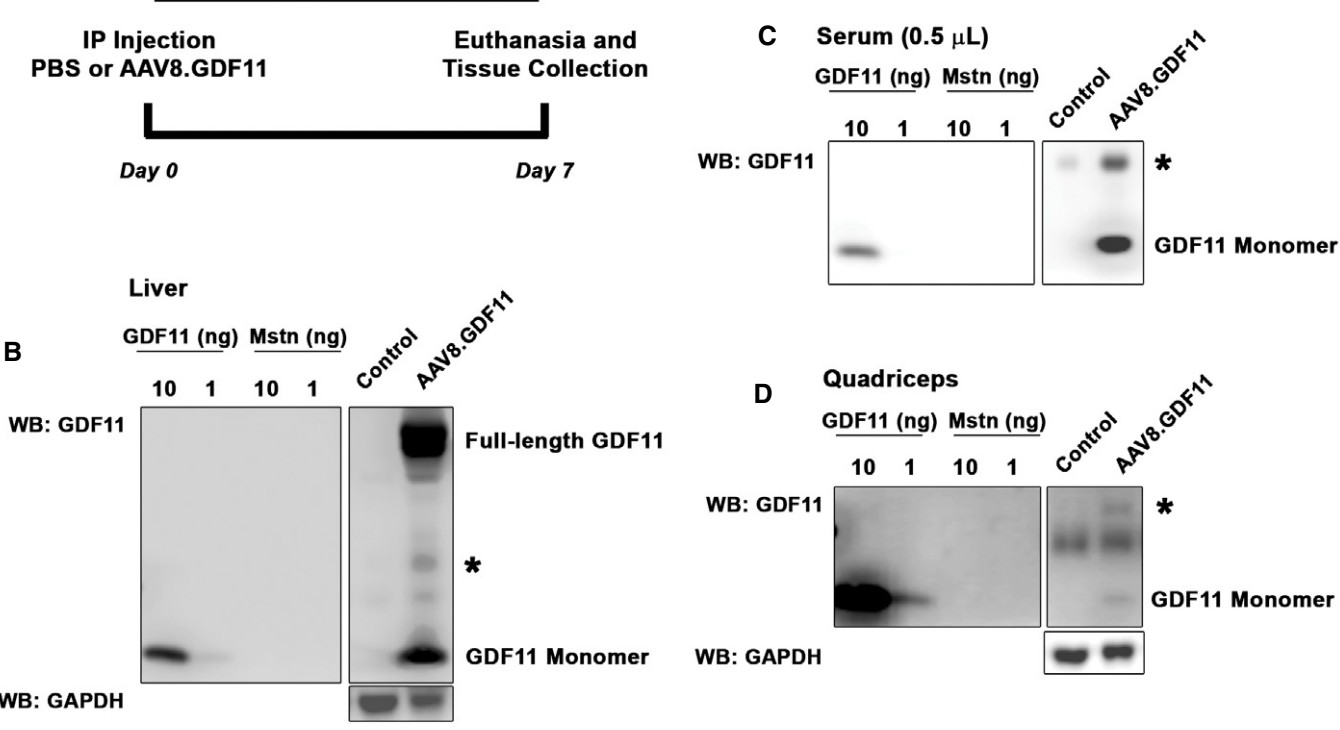

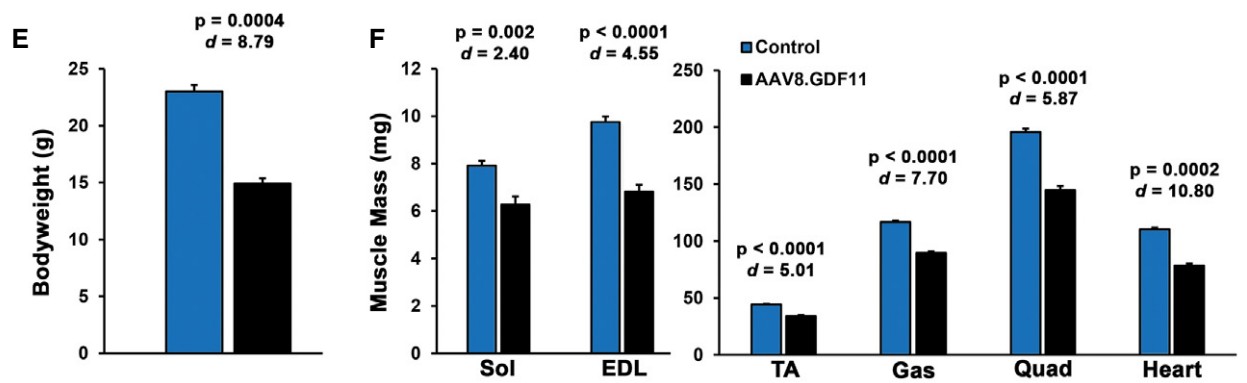

**Figure 3.  Supraphysiological levels of GDF11 cause atrophy of striated muscle *in vivo*.**

A    Twelve-week-old male C57BL/6 mice (*n* = 3) were injected i.p. with PBS (control) or $1 \times 10^{12}$ gc of a liver-specific GDF11 packaged in AAV2/8 (AAV8.GDF11), and were euthanized seven days after treatment.

B–D    Immunoblotting of IgG-reduced samples (see Materials and Methods) reveals an increase in GDF11 content in AAV8.GDF11-treated liver (B), serum (C), and quadriceps (D), as detected by the R&D Systems GDF11 mouse mAb under reducing (50 mM DTT) conditions. The identifiable bands of full-length GDF11 and monomeric GDF11 in the immunoblots are labeled, while an ambiguous 25 kDa band is marked with a star (\*). GAPDH immunoblotting is shown to demonstrate equal loading among lanes.

E, F    Seven days after, AAV8.GDF11 treatment resulted in substantial losses in body weight (E), and muscle mass of the soleus (Sol), extensor digitorum longus (EDL), tibialis anterior (TA), gastrocnemius (Gas), quadriceps (Quad), and heart (F). Values depicted are mean ± SEM. Statistical analysis was performed using two-tailed Student's *t*-test with effect size presented as Cohen's *d* (*d*).

decreased by AAV8.GDF11 (Fig 4A and B). Interestingly, long-term experiments involving manipulations of Mstn/GDF11 (Fig EV3) substantially affect skeletal muscle mass, but not heart mass. For instance, a 12-week treatment cohort with an equal dose of AAV8.Mstn demonstrates similar muscle mass decrements (relative to respective controls) as the seven-day AAV8.GDF11 group, though

no effects on cardiac mass are seen (Fig EV3A). Additionally, inhibition of either Mstn or GDF11 in the same 12-week trial by liver targeted overexpression of BMP-1/tolloid metalloproteinase-resistant mutations either of Mstn (D76A) or GDF11 (D120A) propeptides (dnMstn and dnGDF11, respectively) results in substantial increases in muscle mass without affecting heart mass

## A Skeletal Muscle

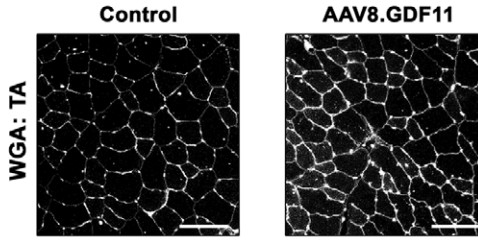

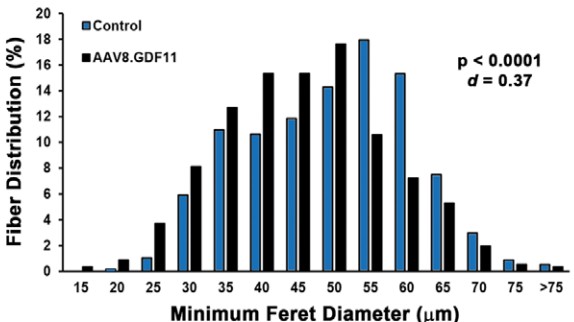

## B Cardiac Muscle

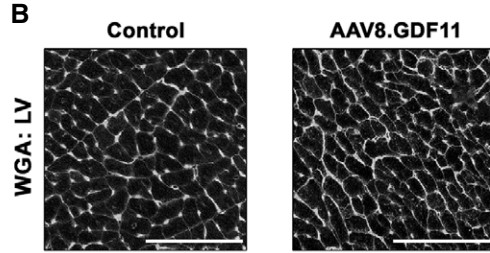

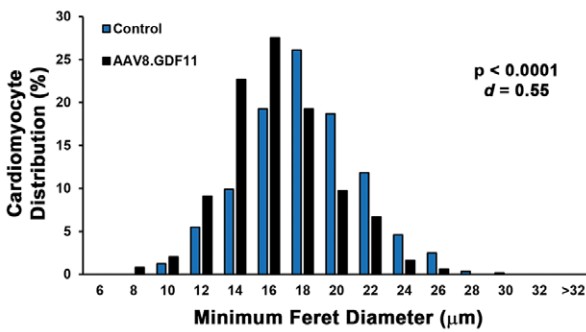

**Figure 4. Myofiber and cardiomyocyte atrophy induced by GDF11.**

A, B Wheat germ agglutinin (WGA; Texas Red-conjugated) stained sections of the tibialis anterior (TA; A) and left ventricle (LV; B) from PBS (control) and liver-specific GDF11 (AAV8.GDF11)-treated mice (*n* = 3) were evaluated for minimum Feret diameter of the individual myocytes (*n* = 500–600 cells/group). The scale bars represent 100 μm. Myocyte size data are presented as a histogram of minimum Feret diameter distribution. Statistical analysis was performed using two-tailed Student's *t*-test with effect size presented as Cohen's *d* (*d*).

(Fig EV3A). Likewise, Mstn knockout mice demonstrate this same trend (Cohn *et al*, 2007) (Fig EV3B). These data demonstrate that GDF11 causes atrophy in both skeletal and cardiac muscle, but only skeletal muscle atrophy is seen by overexpression of Mstn.

Consistent with previous reports (Egerman *et al*, 2015) and the *in vitro* reporter assays described above, systemic elevation by AAV8.GDF11 induced strong phosphorylation of SMAD3 in quadriceps muscle (Fig 5A), suggesting the atrophic effects of GDF11 in skeletal muscle involve the canonical signaling pathway. GDF11 also showed a negative physiological effect on the opposing p-SMAD1/5/8 pathway (Fig EV4A), while SMAD4 content is variable upon GDF11 stimulation (Fig EV4A). GDF11 also increases Akt content without consistently affecting p-Akt (Fig 5A), which may be a compensatory response to the atrophy. The systemic overexpression of GDF11 also affects the phosphorylation of p38 MAPK (Fig 5A), suggesting non-canonical signaling may play a secondary role to the strongly elevated canonical pathway in skeletal muscle. NOX4 content, which was recently shown to have a major role in TGFβ-mediated muscle dysfunction (Waning *et al*, 2015), does not appear to be affected by GDF11 (Fig EV4A).

In the heart, the p-SMAD3 response to AAV8.GDF11 is not as prominent as in skeletal muscle at 7 days following treatment, which coincides with substantial loss of SMAD3 protein in the AAV8.GDF11 group (Fig 5B). Elevated phosphorylation of SMAD3 in the heart is seen at the earlier time points of 3 and 5 days post-treatment (Fig 5C), suggesting that cardiac muscle may differ at regulating SMAD3 signaling following chronic stimulation than skeletal muscle. As in skeletal muscle, the SMAD1/5/8 pathway is decreased, while SMAD4 content is unchanged (Fig EV4B). Contrary to skeletal muscle, non-canonical signaling, as evidenced by large increases in the phosphorylation of p38 MAPK and ERK1/2 (Fig 5B), is more highly elevated than the canonical pathway. This activation does not appear to be mediated through stimulation of TGFβ-activated kinase (TAK) 1 signaling (Fig EV4C).

To determine whether this cachexic phenotype is accompanied by increased expression of muscle "atrogenes", gene expression of the muscle-specific E3 ubiquitin ligases *Fbxo32* (MAFbx/atrogin-1 gene), *Trim63* (MuRF1 gene), and *Fbxo30* (MUSA1 gene) (Bodine *et al*, 2001; Sartori *et al*, 2013) was measured in the quadriceps and hearts of mice treated with AAV8.GDF11 for 3 and 5 days (Fig 5D). At 5 days of exposure, modest ~2.5-fold upregulations of *Fbxo32* and *Trim63* were found in the quadriceps, while *Fbxo30* expression remained unchanged. *Trim63* expression in the heart was elevated ~twofold at both days 3 and 5, while *Fbxo32* expression was unchanged and *Fbxo30* expression became highly variable. These data suggest the striated muscle "atrogene" program is activated by systemic GDF11 overexpression; however, this activation is very modest in comparison with those shown by other atrophy models (Bodine *et al*, 2001; Sacheck *et al*, 2007).

### Pharmacological levels of full-length GDF11 uniquely induce cachexia and cardiac atrophy

To determine whether exposure levels similar to those reported to have pharmacological benefit also induce such a degree of muscle atrophy, a dose de-escalation study was performed using $1 \times 10^{11}$ gc (hereafter referred to as "mid dose") and $5 \times 10^{10}$ gc (low dose) of AAV8.GDF11 in 12-week-old mice (Fig EV5A). Similar to the high

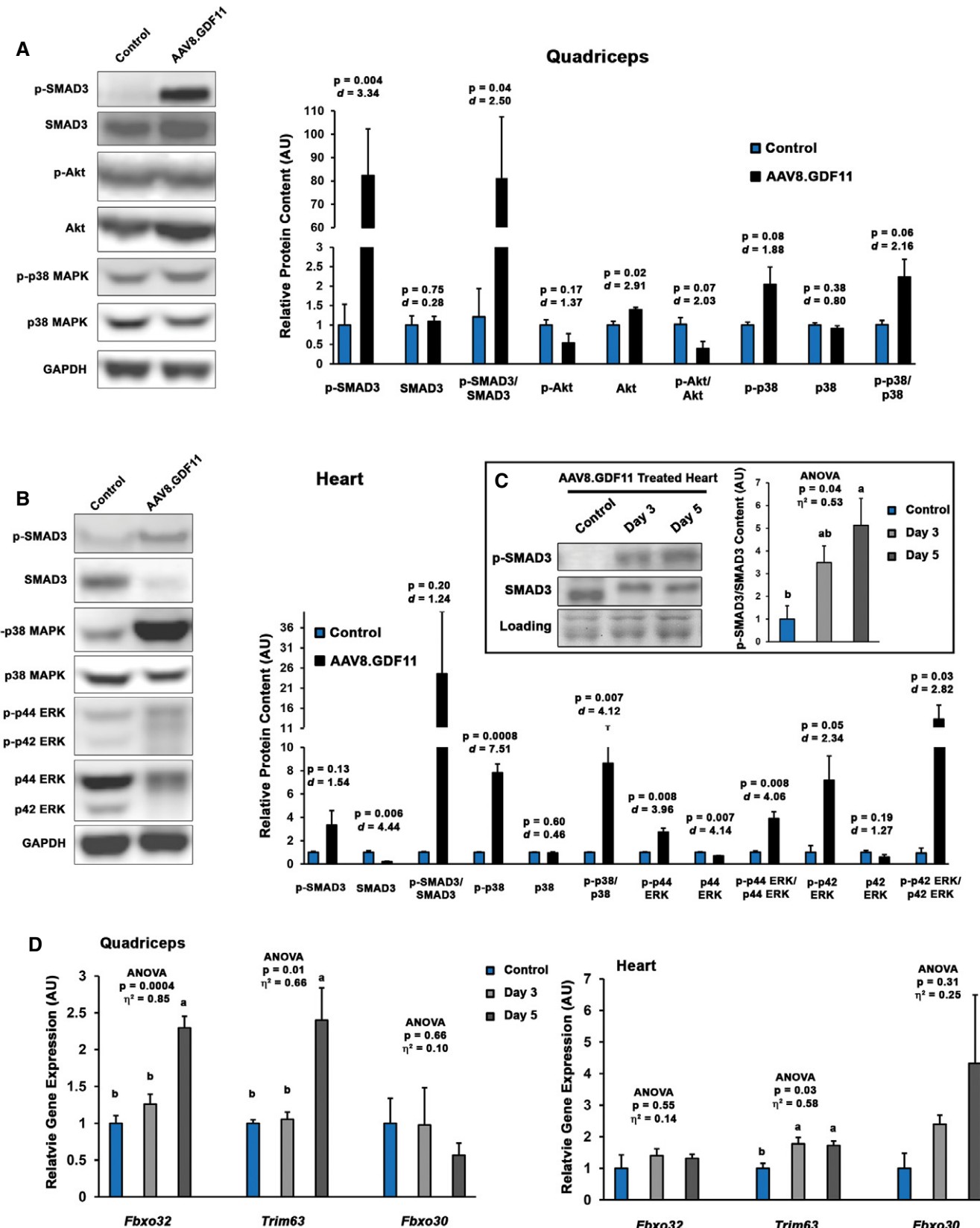

**Figure 5.**

◄

> **Figure 5.  GDF11-induced signaling and atrogene expression in skeletal and cardiac muscle.**
>
> A   Immunoblotting data from quadriceps of PBS (control; *n* = 3) and liver-specific GDF11 (AAV8.GDF11; *n* = 3)-treated male C57BL/6 mice for phosphorylated and total forms of SMAD3, Akt, and p38 MAPK. Loading is normalized by GAPDH content and quantified relative to control values.
>
> B   Immunoblotting data for phosphorylated and total forms of SMAD3, p38 MAPK, and p42/p44 ERK in the hearts of control and AAV8.GDF11-treated male C57BL/6 mice. Loading is normalized by GAPDH content and quantified relative to control values.
>
> C   Phosphorylation status of SMAD3 in the hearts of control (*n* = 3) and AAV8.GDF11-treated male C57BL/6 mice at 3 days (*n* = 4) and 5 days (*n* = 4) following injection. Loading is normalized by Ponceau Red staining and quantified relative to control values.
>
> D   Gene expression of *Fbxo32* (MAFbx gene), *Trim63* (MuRF1 gene), and *Fbxo30* (MUSA-1 gene) in the quadriceps (left) and heart (right) of control (*n* = 3) and AAV8.GDF11-treated male C57BL/6 mice at 3 days (*n* = 4) and 5 days (*n* = 4) following injection. Relative gene expression values were calculated by the ΔΔC$_t$ method using *Gapdh* as the reference gene.
>
> Data information: Values depicted are mean ± SEM. In (A, B), statistical analysis was performed using two-tailed Student's *t*-test with effect size presented as Cohen's *d* (*d*). In (C, D), statistical analysis was performed using one-way ANOVA analysis with Tukey's HSD *post hoc* test (non-connecting letters indicate $P < 0.05$ between groups) and effect size presented as eta-squared ($\eta^2$).

dose ($1 \times 10^{12}$ gc; described above), AAV-mediated GDF11 expression was evident within 10 days post-injection (Fig EV5B), resulting in estimated levels of 0.03–0.09 mg/kg and 0.01–0.02 mg/kg for the mid- and low-dose groups, respectively (based on the above-mentioned calculations for a 23 g mouse from serum levels of 0.5–1.6 ng/μl and 0.26–0.34 ng/μl, respectively). These levels are below those administered in recent pharmacological studies. The study was terminated at this 10-day time point due to substantial wasting (~25% loss of body weight) and mobility issues from the mid dose, requiring immediate euthanasia (Fig EV5C). At this time point, the low-dose group displayed ~10% loss of body weight, though no overt pathology was evident. While mid-dose mice displayed both skeletal and cardiac muscle atrophy at this time point, the low-dose group only began to exhibit a cardiac phenotype (Fig EV5D).

From the above data, we hypothesized that Mstn and GDF11 would induce similar phenotypes if expressed at similar levels. As the low dose of AAV8.GDF11 and a high dose ($2 \times 10^{12}$ gc) of AAV8.Mstn demonstrate comparable serum exposures at 10 days post-injection (Fig 6A), we tested this hypothesis in 7-week-old male C57BL/6 mice (experiment depicted in Fig 6B). As displayed in Fig 6C, both Mstn and GDF11 groups lost body mass after treatment, while control mice continued to grow. The GDF11 group began exhibiting signs of severe cachexia beginning at day 14, and the study was terminated following the death of a mouse at day 16 (21% loss of initial body mass by the group). While the Mstn-treated mice lost ~8% of their initial body mass at this time point, they otherwise appeared healthy and demonstrated normal activity.

At time of euthanasia, there were no significant differences in absolute body weight or skeletal muscle mass between the Mstn and GDF11 groups (Fig 6D–F); however, the hearts of Mstn-treated mice did not show the atrophic effects seen in those treated with GDF11. Even when normalized to either body weight or tibia length, Mstn-treated hearts were larger than those with GDF11 treatment (Fig 6G). Stomach contents were comparable between the treatment groups, suggesting anorexia was not responsible for this wasting phenotype. Likewise, the observed differences in mass are due to atrophy of the tissues, not failure to grow, as the values for 7-week-old mice from this colony are similar to control treatment values. Interestingly, both Mstn- and GDF11-affected liver mass, while only GDF11 atrophied the kidneys (Fig EV5E). These divergent effects of the ligands on heart and kidney mass are not due to differential accumulation of GDF11 and Mstn in the tissue (Fig EV5F), which appears, as one would expect, to be proportional to tissue vascularity.

### Receptor expression patterns differ between skeletal and cardiac muscle

As Mstn and GDF11 affect skeletal muscle mass similarly and only GDF11 affects the heart, it is possible differential receptor utilization may occur in a tissue-specific context, resulting in divergent effects of the ligands on cardiomyocytes at the levels expressed in this study. We thus investigated associated receptor levels/expression in skeletal muscle and heart. The heart contains nearly twofold the content of the common type II receptor, ActRIIB, as quadriceps, which was unchanged in either tissue by ligand manipulation (Fig 7A and B). Though gene expression of *Acvr1b* (ALK4 gene) is comparable between heart and quadriceps (Fig 7C), the expression of *Tgfbr1* (ALK5 gene) is nearly twofold higher in the heart (Fig 7D). When normalized to *Acvr1b*, the heart contains ~60% more *Tgfbr1* than the quadriceps (Fig 7E), which suggests that signaling induced by either GDF11 or Mstn in the heart is more likely to be mediated by ALK5 than ALK4, the primary mediator of Mstn signaling in myoblasts (Kemaladewi *et al*, 2012) and possibly mature muscle. At these expression levels, it is possible that GDF11 binding to ActRIIB may preferentially recruit ALK5 more so than Mstn, explaining the differential effects of the two ligands in the heart. The heightened *Tgfbr1* expression in the heart is likely linked to the 5.5-fold higher gene expression of *Tgfb1* in the heart than skeletal muscle (Fig 7F), as TGFβ signals through dimerization of ALK5 with TGFβ type II receptor (TβRII) and is a positive regulator of cardiomyocyte size (Rosenkranz, 2004). Thus, differential receptor profiles between skeletal and cardiac muscle offer a potential explanation for the potent effects of GDF11 on cardiac mass in comparison with Mstn. Marked elevation of *Tgfb1* expression in the heart following 3 and 5 days of exposure to high-dose AAV8.GDF11 (Fig 7G) supports this hypothesis, as it appears the heart is upregulating *Tgfb1* to compensate for the loss of cardiomyocyte mass.

## Discussion

Given the recent controversy surrounding GDF11 as an anti-aging therapy (Harper *et al*, 2016; Walker *et al*, 2016), it is clear that additional mechanistic studies on this TGFβ superfamily member are required. While the physiological role of endogenous GDF11 in postnatal striated muscle remains ambiguous, the biomedical importance of this ligand is exemplified by the recent demonstration that

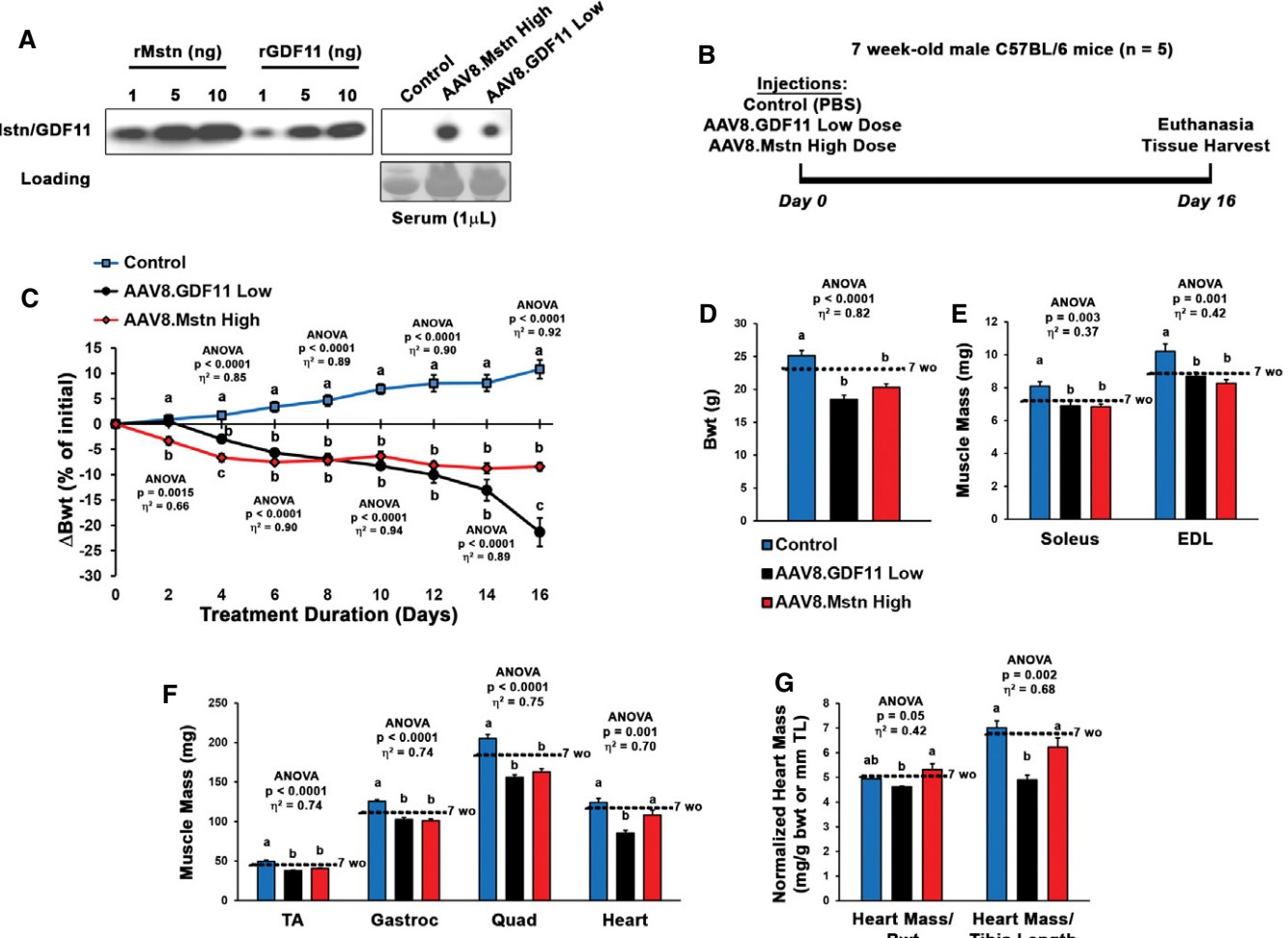

**Figure 6. Myostatin and GDF11 have differential effects *in vivo*.**

A    Nearly equivalent serum exposure of myostatin (Mstn) and GDF11 can be obtained by treatment with $2 \times 10^{12}$ gc of AAV8.Mstn (high dose) or $5 \times 10^{10}$ gc of AAV8.GDF11 (low dose) 10 days following injection.

B, C    Seven-week-old C57BL/6 male mice were treated with PBS (control; $n = 5$), AAV8.GDF11 low dose ($n = 5$), or AAV8.Mstn high dose ($n = 5$) and monitored for 16 days until experiment was terminated due to the death of an AAV8.GDF11-treated mouse to severe cachexia (depicted in B). The change in body weight (Bwt) in these groups across the 16 days is displayed in (C).

D–G    Morphological measurements of surviving mice (thus $n = 4$ for AAV8.GDF11 group) collected at tissue harvest, including Bwt (D), muscle mass of soleus and extensor digitorum longus (EDL), tibialis anterior (TA), gastrocnemius (Gastroc), quadriceps (Quad), and heart (E–F). Heart mass normalized to both Bwt (in g) and tibia length (TL; in mm; G). Mean values for 7-week-old mice from this colony ($n = 5$) are indicated by the dotted lines to show starting masses.

Data information: Values are displayed as mean ± SEM. Statistical analysis performed using one-way ANOVA analysis with Tukey's HSD *post hoc* test (non-connecting letters indicate $P < 0.05$ between groups) and effect size presented as eta-squared ($\eta^2$) for ANOVA analyses. Note that "a" on Day 2 of panel (C) refers to the 2 overlapping groups.

elevated GDF11 levels associate with patient frailty and problematic recovery following cardiovascular disease (CVD) intervention (Schafer *et al*, 2016). In the current report, we demonstrate that recombinant GDF11 and Mstn similarly act through the canonical SMAD2/3 pathway and reduce myotube diameter *in vitro*. Additionally, we provide evidence that overexpression of full-length, bioactive GDF11 is a powerful inducer of atrophy in both skeletal and cardiac muscle and induces a severe cachexic phenotype *in vivo*. Thus, it is not likely a candidate for an "anti-aging" therapeutic.

TGFβ superfamily ligands can have substantial influence on skeletal muscle mass. In addition to the muscle effects of Mstn

(described and demonstrated above), activin A, also an ActRIIB ligand, induces muscle atrophy (Chen *et al*, 2014) and has been implicated (along with Mstn) in cancer-induced muscle wasting (Zhou *et al*, 2010). Soluble ActRIIB or anti-ActRIIB antibodies, which can block activity of Mstn, GDF11, and activin A, substantially increase muscle mass and improve muscle wasting phenotypes (Morine *et al*, 2010b; Zhou *et al*, 2010). TGFβ1, signaling through ALK5 and TβRII to activate SMAD2/3, also induces a muscle atrophy phenotype (Mendias *et al*, 2012; Narola *et al*, 2013). Conversely, activation of the SMAD1/5/8 pathway via BMP receptor signaling promotes muscle hypertrophy (Sartori *et al*,

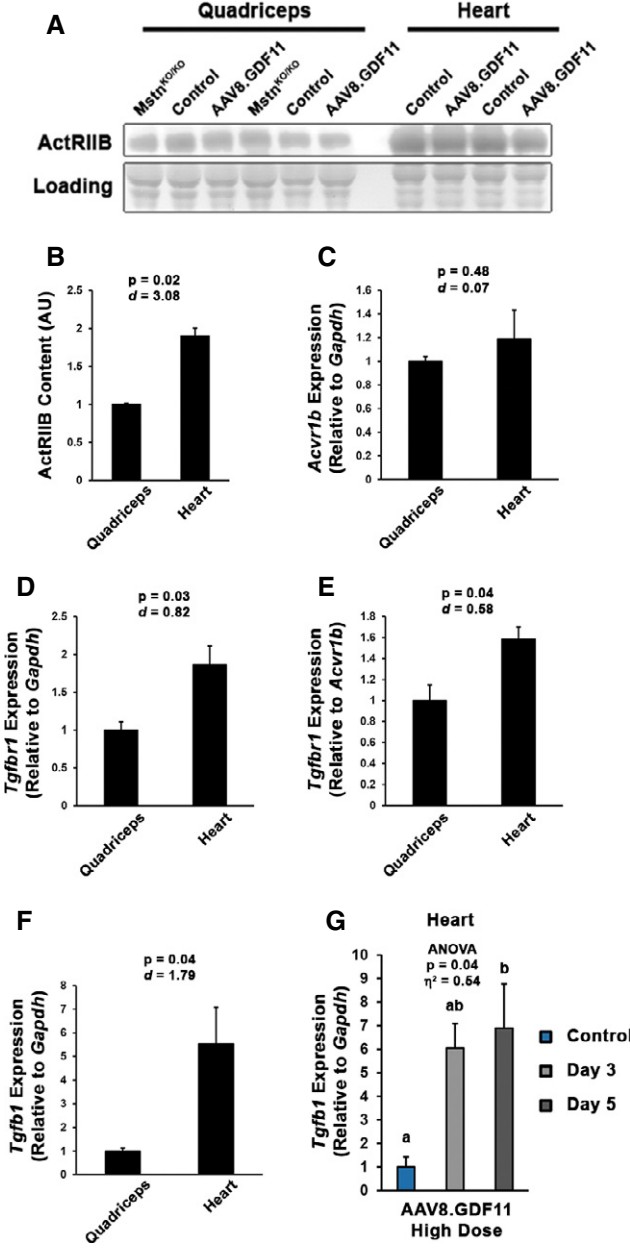

**Figure 7. Differential activin receptor levels in skeletal muscle and heart.**

A, B Relative muscle content of the activin type IIB receptor (ActRIIB) in quadriceps ($n = 6$) and hearts ($n = 4$) from multiple treatment groups, as determined by immunoblotting (normalized to Ponceau-visualized loading).

C–F Gene expression of *Acvr1b* (ALK4 gene; C), *Tgfbr1* (ALK5 gene; D and E), and *Tgfb1* (F) in quadriceps and heart of untreated 7-week-old C57BL/6 mice ($n = 3$), as measured by real-time PCR. Relative gene expression values were calculated by the $\Delta\Delta C_t$ method using *Gapdh* (C, D and F) or *Acvr1b* (E) as reference genes.

G Cardiac gene expression of *Tgfb1* in control ($n = 3$), day 3 ($n = 4$), and day 5 ($n = 4$) high-dose ($1 \times 10^{12}$ gc) AAV8.GDF11-treated C57BL/6 mice. Relative gene expression values were calculated by the $\Delta\Delta C_t$ method using *Gapdh* as the reference gene.

Data information: Values are displayed as mean $\pm$ SEM. In (B–F), statistical analysis performed using two-tailed Student's *t*-test, with effect size presented as Cohen's *d* (*d*). In (G), statistical analysis performed using one-way ANOVA with Tukey's HSD *post hoc* test (non-connecting letters indicate $P < 0.05$ between groups) and effect size presented as eta-squared ($\eta^2$).

2013; Winbanks *et al*, 2013). It is therefore not surprising that GDF11, which similarly ligates to ActRIIB and activates SMAD2/3 as Mstn, induces skeletal muscle atrophy *in vitro* and *in vivo*.

In the current study, an estimated dose of 0.01 mg/kg (based on 0.055 ml/g bwt of blood volume for a 23 g mouse; http://www.inf ormatics.jax.org/mgihome/other/mouse_facts1.shtml) of bioactive GDF11 is enough to induce substantial cachexia in male C57BL/6 mice in < 3 weeks of treatment. As none of the recent studies involving rGDF11 treatments (ranging from 0.1 mg/kg; Loffredo *et al*, 2013; Sinha *et al*, 2014; Egerman *et al*, 2015; Smith *et al*, 2015; to 1 mg/kg; Poggioli *et al*, 2016; daily) clearly provide skeletal muscle mass data, it is uncertain whether skeletal muscle atrophy occurred as a result of GDF11 administration in these studies. This is a critical omission for proposals of anti-aging effects of GDF11, as further loss of skeletal muscle mass will severely decrease the quality of life for the elderly and CVD patients, which will result from elevating GDF11. Indeed, the recent report by Schafer *et al* (2016) depicts elevations in endogenous GDF11 as being major detriment to patients with co-morbidities. While it is possible that the ligands used in the aforementioned studies (i.e., mature GDF11 dimer produced in bacteria) are either less potent or have different biological effects as those produced *in vivo*, thus contributing to the discrepancy in reported results, *in vitro* data reported in this work suggest the recombinant protein does display high, ActRIIB-dependent potency when administered in cell culture.

An interesting and unexpected finding of the current work is the increased susceptibility of cardiac muscle to GDF11-induced atrophy. While systemic manipulation (whether Mstn or dnMstn) and genetic deletion of Mstn display minimal effects in the heart despite dramatic changes to skeletal muscle, cardiac-specific expression of Mstn or dnMstn does affect cardiomyocyte size (Bish *et al*, 2010; Heineke *et al*, 2010). This demonstrates Mstn can affect cardiac mass and suggests Mstn is more effective in autocrine fashion in the heart, while systemic levels (even at the levels expressed in the current study) are inadequate to affect cardiac muscle size. We propose a possible explanation for this phenomenon to be differential receptor profiles and utilization between heart and skeletal muscle, as evidenced by increased ActRIIB content and *Tgfbr1*: *Acvr1b* expression in the heart. As ALK4 is the primary mediator of Mstn signaling in myoblasts (Kemaladewi *et al*, 2012), is it reasonable to suspect that Mstn binding to ActRIIB preferentially recruits ALK4 as the type I receptor to initiate signaling. Likewise, GDF11 binding may preferentially recruit ALK5, explaining the differential effects of the two ligands in the heart. Additionally, the heart expresses more *Tgfb1* than skeletal muscle; therefore, it is also possible that the atrophic effects of GDF11 of on the heart could additionally involve ActRIIB out-competing TβRII for ALK5. We cannot, however, rule out the possibility of signaling modulators, such as the co-receptor cripto (Kemaladewi *et al*, 2012), or receptor post-translational modifications, such as SUMOylation (Miyazono *et al*, 2008), being involved in these divergent effects of Mstn and GDF11 on cardiac muscle, as well.

Another element of the existing GDF11 debate concerns proper reagents for the detection of GDF11 (Egerman *et al*, 2015; Rodgers & Eldridge, 2015; Poggioli *et al*, 2016). As shown by Egerman *et al* (2015), both the SOMAmer and the anti-GDF11 antibody (Abcam #ab124721; now labeled as anti-GDF8/11) used in the initial report by Loffredo *et al* (2013) detect Mstn and GDF11 to similar extents.

In this study, an assay utilizing the GDF11-specific clone 743833 [R&D Systems; specificity replicated previously (Smith *et al*, 2015) and in the current report] suggests GDF11 immunoreactivity is increased with age in both rat and human serum. In the present study, we used antibodies verified to be specific for both full-length and monomeric GDF11 and Mstn (clone 743833 and REGN459, respectively; Fig EV2), allowing us to reliably detect either species in IgG-reduced tissue lysate and serum.

As Poggioli *et al* (2016) attribute some of the immunoreactivity of ab124721 to be IgG light chains (hence appearing as Mstn/GDF11 dimer at ~25 kDa), we find an ambiguous 25 kDa anti-mouse IgG immunoreactive band detected in AAV8.GDF11-treated samples, which cannot be removed with pre-treatment of samples with protein A/G and/or protein L-coated agarose beads (Fig EV2D). Furthermore, purified rGDF11 does not react with anti-mouse IgG in either reduced or non-reduced forms (Fig EV2D). These observations suggest the detection of dimeric GDF11 from biological samples (both reducing and non-reducing) may be further complicated by anti-mouse IgG immunoreactive species that appear to be induced by increasing GDF11 levels, and is perhaps the same species detected by Abcam's ab124721.

With the current debate over the therapeutic potential of GDF11 for age-related phenomena, a greater understanding of the role of this protein in physiological and pathological processes is needed. In the current report, it is shown that GDF11 is a potent negative regulator of striated muscle mass when expressed at supraphysiological levels. While it is still unclear what role, if any, that GDF11 plays in normal striated muscle physiology, it clearly activates similar pathways as its close relative, Mstn, and can induce pronounced muscle wasting and cachexia when elevated systemically. In fact, the degree of atrophy in skeletal and cardiac muscle demonstrated by this ligand is very reminiscent of that found in a severe murine model of cancer cachexia, which is ameliorated by anti-ActRIIB treatment (Zhou *et al*, 2010). Interestingly, GDF11 has been suggested to be expressed by human cancers (Yokoe *et al*, 2007). This suggests caution should be used when considering the therapeutic potential of GDF11, as the presently described data suggest its true biological actions would greatly exacerbate muscle loss in already susceptible populations. In light of recent reports (Schafer *et al*, 2016) and the currently described data, we suggest that the biomedical focus of GDF11 to shift away from therapeutic potential and toward its potential role as a pathological effector/biomarker, particularly in conditions of muscle wasting.

# Materials and Methods

### AlphaLISA assays

To assess the phosphorylation of SMAD in response to TGFβ super-family members in undifferentiated cells, C2C12 myoblasts (ATTC CRL-1772; passage 14) and HEK293T cells (ATCC CRL-11268; passage 13) were plated on 96-well tissue culture-treated plates (Greiner-Bio-One 655098) in growth medium [high glucose Dulbecco's modified Eagle's medium (DMEM) + 10% fetal bovine serum (FBS) + 1% penicillin streptomycin (PS)], grown to confluency, and were treated with various doses of the following recombinant ligands for 1 h ($n = 4$): Mstn (R&D Systems #788-G8), GDF11

(R&D Systems # 1958-GD), TGFβ (R&D Systems #240-B), activin A (R&D Systems # 338-AC), BMP-2 (R&D Systems # 355-BM), BMP-4 (R&D Systems # 314-BP), BMP-6 (R&D Systems # 507-BP), or BMP-7 (R&D Systems # 354-BP). All recombinant proteins were reconstituted according to the manufacturer's instructions. Cells were lysed using Phosphosafe Buffer (VWR EM71296-4) for 45 min, and phosphorylation of SMAD2/3 or SMAD1/5/8 was determined using AlphaLISA Surefire Ultra kits (PerkinElmer ALSU-PSM3 or ALSU-PSM1, respectively) read on a PerkinElmer EnVision 2103 Multilabel reader. The Alpha signal was normalized to the maximum signal measured for that cell type and readout (either p-SMAD2/3 or p-SMAD1/5/8), so that relative contributions of each ligand to the p-SMAD signal could be evaluated.

To assay ligand response in differentiated myotubes, C2C12 myoblasts were plated on in growth medium until confluency (~24 h), after which were switched to differentiation medium (DMEM + 2% horse serum + 1% PS) to induce differentiation to myotubes (day 0). Media was exchanged on day 2. At day 6, myotube cultures were treated with various doses of the recombinant ligands (listed above) for 1 h ($n = 4$), and SMAD phosphorylation was assayed as mentioned above. For ActRIIB antibody (anti-ActRIIB) assays, HEK293T cells were treated with 0 or 100 nM anti-ActRIIB (Creative BioMart, murinized version of BYM338 antibody; Lach-Trifilieff *et al*, 2014) overnight prior to ligand addition, lysing, and AlphaLISA evaluation, as described above.

### Myotube diameter assay

C2C12 myoblasts (ATTC CRL-1772; maintained between passages 3 and 7) were cultured in growth medium until confluency, after which were switched to differentiation medium (low glucose DMEM + 2% FBS + 1% PS) to induce differentiation to myotubes (day 0, as depicted in Fig 2A). At day 7 of differentiation, myotube cultures were separated into control ($n = 5$), 50 ng/ml recombinant Mstn (R&D Systems #788-G8; $n = 4$), 50 ng/ml recombinant GDF11 (R&D Systems #1958-GD; $n = 5$), or 50 ng/ml recombinant TGFβ (R&D Systems #240-B; $n = 3$) enriched differentiation media for 72 h. All recombinant proteins were reconstituted according to the manufacturer's instruction in 4 mM HCl with 0.1% bovine serum albumin (BSA).

At day 10, cultures were fixed in 4% paraformaldehyde (PFA), permeabilized in 0.1% Triton X-100, and blocked in 0.2% BSA for 1 h. Myotubes were stained with anti-myosin heavy chain (clone MY-32; Sigma-Aldrich #M1570; St. Louis, MO) overnight, incubated with Alexa-568-conjugated goat anti-mouse secondary antibody (Life Technologies #A11031; Grand Island, NY) for 1 h, and counter stained with DAPI (Sigma-Aldrich). Images were acquired with a Leica TSC-SP8 confocal microscope and analyzed by Leica LAS X software (200–400 myotubes per condition).

### Animals

All procedures and experiments were approved by and conducted in accordance of the University of Pennsylvania IACUC and the University of Florida IACUC. This study used male C57BL/6J mice from colonies originally obtained from Jackson Laboratory (Stock #000664). Mstn^KO/KO, wild-type, and heterozygous littermates are congenic on the C57BL/6 background. Mice were housed 3–5 mice per cage, randomly assigned into treatment groups, provided

*ad libitum* access to food and water with enrichment, and maintained on a 12-h light/dark system. Following allotted time, mice were euthanized. Muscles were quickly dissected, weighed, and either snap-frozen in liquid $N_2$ or embedded in OCT and frozen in melting isopentane. The apical half of the heart was snap-frozen, while the rest was frozen in OCT. Frozen tissue was stored at −80°C until analysis. Sample sizes were determined based on power analyses of primary measures from previous studies. Mice were randomized prior to treatment protocols.

## Adeno-associated virus production and treatment

Codon optimized cDNA encoding full-length mouse GDF11 (NP_034402) and the D120A GDF11 propeptide (Ge *et al*, 2005; dnGDF11) were synthesized (Integrated DNA Technologies; Coralville, IA). These constructs, full-length Mstn, and the D76A Mstn propeptide (Morine *et al*, 2010a) (dnMstn) were cloned behind the liver-specific $\alpha_1$-antitrypsin promoter with ApoE enhancer in the pLSP AAV shuttle vector (Morine *et al*, 2010a), and were packaged into adeno-associated virus (AAV) pseudotype 2/8 by the University of Pennsylvania vector core. Mice were injected i.p. with the specified genome content (gc) of virus diluted to 50 µl in sterile PBS. Control animals received 50 µl of PBS.

## Immunoblotting

Snap-frozen samples were finely crushed and homogenized in T-PER buffer (Thermo Scientific; Waltham, MA) supplemented with protease and phosphatase inhibitors (Thermo Scientific). Protein concentration of resulting supernatant was determined using Bio-Rad Protein Assay (Bio-Rad; Hercules, CA). Serum samples were diluted to the indicated concentration in PBS. Endogenous mouse IgGs were reduced from samples intended for analysis with mouse monoclonal antibodies by two 1-h incubations with protein A/G-coated agarose beads (Santa Cruz Biotechnology #sc-2003; Dallas, TX; pre-cleaned with homogenization buffer) at 4°C. Protein L-coated agarose beads (Santa Cruz #sc-2336) were used where indicated to remove IgG light chains. Lysates were removed from beads following each incubation by centrifugation at 1,000 *g* and removing the supernatant.

Protein samples were boiled for 10 min in 4× sample buffer containing 50 mM DTT (unless under non-reducing conditions), subjected to SDS–PAGE using 4–12% SDS–polyacrylamide gels (Life Technologies), and transferred to nitrocellulose membranes using the iBlot system (Life Technologies). Unless otherwise noted, 50 µg of tissue homogenates was loaded per lane. Membranes were blocked in 5% milk-TBST or 5% BSA-TBST, and incubated with primary antibody overnight at 4°C. Following TBST washes, membranes were incubated in the appropriate HRP-conjugated anti-rabbit (Cell Signaling #7074; Danvers, MA) or anti-mouse (Cell Signaling #7076) secondary antibody for 1 h at room temperature, washed, incubated for 5 min in ECL reagent (Thermo Scientific), and imaged using the LI-COR C-DiGit (LI-COR Biosciences; Lincoln, NE) imaging system. Blots for phosphorylated proteins were stripped and re-probed for total protein, unless initial signal was too strong to be reliably removed by stripping buffer. All membranes underwent a final probe for GAPDH (Santa Cruz Biotechnology #sc-25778) and were stained with Ponceau Red for equal loading

### The paper explained

#### Problem

The transforming growth factor β superfamily of signaling ligands can have substantial influence on striated muscle mass. Growth and differentiation (GDF) 11, a member of this family, has been purported as a therapeutic for striated muscle decrements; however, the effects of this molecule on muscle mass have not been described despite high homology to myostatin (Mstn), a potent stimulator of muscle atrophy. The objective of the current report was to delineate the effects of bioactive GDF11 on striated muscle mass.

#### Results

In cell culture experiments, we found that GDF11 and Mstn have similar signaling and atrophy-inducing activities in skeletal myocytes. Using adeno-associated virus to induce systemic overexpression of GDF11 in mice, substantial atrophy of skeletal and cardiac muscle was seen, resulting in a lethal wasting phenotype which did not occur in mice expressing similar levels of Mstn.

#### Impact

These data indicate that bioactive GDF11 at supraphysiological levels are not beneficial to striated muscle, causing wasting of both skeletal and cardiac muscle. Rather than a therapeutic agent, GDF11 is a deleterious biomarker involved in muscle wasting.

verification. Band signal intensities were measured using Image Studio Lite software (LI-COR Biosciences), normalized to sample loading, and reported relative to respective control samples. Primary antibodies used for this study include the following: GDF11 (R&D Systems #MAB19581; Minneapolis, MN), Mstn (C-terminal; kind gift from Regeneron Pharmaceuticals), p-SMAD2 (S465/467; Cell Signaling #3108), SMAD2 (Cell Signaling #5339), p-SMAD3 (S423/425; Cell Signaling #9520), SMAD3 (Cell Signaling #9523), p-SMAD1/5/8 (S463/465; Cell Signaling #9511), SMAD5 (Cell Signaling #12534), SMAD4 (Cell Signaling #9515), p-Akt (S473; Cell Signaling #9271), Akt (Cell Signaling #9272), p-p38 MAPK (T180/Y182; Cell Signaling #9211), p38 MAPK (Cell Signaling #9212), p-ERK1/2 (T204/Y204; Cell Signaling #9101), ERK1/2 (Cell Signaling #9102), NOX4 (Abcam #ab133303; Cambridge, MA), p-TAK1 (T184/187; Cell Signaling #4531), TAK1 (Cell Signaling #4505), ActRIIB (Sigma-Aldrich #A0457), and Mstn/GDF11 (Abcam #ab124721).

## Muscle fiber and cardiomyocyte size

OCT-embedded tibialis anterior and heart samples were sectioned at 10 µm, fixed in ice-cold acetone for 10 min, and stained with Texas Red-conjugated WGA (Life Technologies) diluted 1:500 in PBS for 1 h. Following incubation, slides were washed with PBS, coversliped with Vectashield (Vector Labs; Burlingame, CA) mounting medium, and imaged using a Leitz DMRBE microscope equipped with a Leica DCF480 digital camera. Myocyte size analysis ($n$ = 500–600 cells/group) was performed using ImageJ software by investigators blind to experimental design, and recorded as minimum Feret diameter in µm.

## Real-time PCR

RNA was isolated from finely crushed snap-frozen mouse quadriceps and heart samples using Trizol Reagent (Life Technologies),

treated with DNAse (Promega; Madison, WI), and reverse transcribed using the SuperScript III kit (Life Technologies). Resulting cDNA was subjected to real-time PCR using RQG SYBR Green supermix (Qiagen) in a Rotor Gene Q real-time PCR machine (Qiagen). The following mouse-specific primers were used: *Fbxo32* (forward) 5′-GCA GCC GCT CAG CAT TCC CA-3′ and (reverse) 5′-ACC GAC GGA CGG GAC GGA TT-3′; *Trim63* (forward) 5′-AGG GCT CCC CAC CAC TGT GT-3′ and (reverse) 5′-TTG CCC CTC TCT AGG CCA CCG-3′; *Fbxo30* (forward) 5′-ATC GAT GGC CGG TTA GTT ATT CA-3′ and (reverse) 5′-GCC CCT ATC TCA CCC TCA TCA AG-3′; *Acvr1b* (forward) 5′-GAA CCG CTA CAC AGT GAC CA-3′ and (reverse) 5′- AAT TCC CGG CTT CCC TTG AG-3′; *Tgfbr1* (forward) 5′-GCA TTG GCA AAG GTC GGT TT-3′ and (reverse) 5′-TGC CTC TCG AAA CCA TGA AC-3′; *Tgfb1* (forward) 5′-GAC TCT CCA CCT GCA AGA CCA T-3′ and (reverse) 5′-GGG ACT GGC GAG CCT TAG TTT-3′; *Gapdh* (forward) 5′-AGC AGG CAT CTG AGG GCC CA-3′ and (reverse) 5′- TGT TGG GGG CCG AGT TGG GA-3′. Relative gene expression quantification was performed using the $\Delta\Delta C_t$ method with either *Gapdh* or *Acvr1b* as the reference gene.

**Statistical analysis**

Statistical analysis was performed using unpaired, two-tailed Student's *t*-test or one-way ANOVA followed by Tukey's HSD *post hoc* test (when ANOVA $P < 0.05$), where appropriate. Effect size is reported as Cohen's *d* ($d$) for *t*-test analyses, and as eta-squared ($\eta^2$) for ANOVA analyses. Values are displayed as mean ± SEM. No data points were excluded from analysis in this study.

**Expanded View** for this article is available online.

## Acknowledgements

This work was funded by a Wellstone Muscular Dystrophy Cooperative Center grant (U54-AR-052646) from the NIH to HLS, Leducq Foundation funding (13CVD04) to HLS, and from funding from the Parent Project Muscular Dystrophy to HLS. The authors acknowledge and greatly appreciate the assistance of Jennifer Pham, Pedro Acosta, Alexandra Agathis, Adam George, Chris Philips, Morgan Venuti, and Cora Coker for their contributions in this study. We also thank Regeneron Pharmaceuticals for generously providing their myostatin C-terminal antibody.

## Author contributions

This study was designed by DWH, MM-B, JYH, JJH, SE, and HLS. Experiments and data collection were performed by DWH, MM-B, JYH, and JJH. Data analysis and interpretation were performed by DWH, MM-B, JYH, and HLS. DWH, JYH, and HLS drafted and revised the manuscript.

## Conflict of interest

JYH, SE, and JJH were employees of Cytokinetics Inc. at the time of this study. The other authors declare that they have no conflict of interest.

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
