## [Review Process File · EMBO Molecular Medicine]

Supraphysiological levels of GDF11 induce striated muscle atrophy

David W. Hammers, Melissa Merscham-Banda, Jennifer Ying Hsiao, Stefan Engst, James J. Hartman, and H. Lee Sweeney

Corresponding author: H. Lee Sweeney, University of Florida College of Medicine

Review timeline:

Submission date:	23 October 2016
Editorial Decision:	09 December 2016
Revision received:	05 January 2017
Editorial Decision:	25 January 2017
Revision received:	26 January 2017
Accepted:	03 February 2017

Transaction Report:

Editor: Roberto Buccione

1st Editorial Decision

09 December 2016

Thank you for the submission of your manuscript to EMBO Molecular Medicine. We are very sorry that it has taken longer than usual to get back to you on your manuscript. We experienced unusual difficulties in securing three willing and appropriate reviewers. Furthermore, we received some of the evaluations with considerable delay.

As you will see, the reviewers clearly find your work both interesting and important. However, while reviewer 1 and 2 are largely positive, reviewer 3 is more reserved.

Specifically, reviewer 3 notes the experimental disconnection between the observations on the effect of GDF11 on the SMAD2/3 pathway and the effects on muscle atrophy and questions whether the observed atrophy actually involves SMAD2/3. S/he also mentions that the explanation provided for differential effect of Mstn and GDF11 on the cardiac muscle is also not documented by the data. Reviewer 3 also suggests that the claim that GDF11-promoted atrophy resembles age-induced atrophy is not sufficiently supported.

Finally, reviewer 1 would like to see whether GDF11-mediate wasting requires the activation of an atrophy programme.

While publication of the paper cannot be considered at this stage, we would be pleased to consider a revised submission, with the understanding that the reviewers' concerns must be addressed including with additional experimental data where appropriate and that acceptance of the manuscript will entail a second round of review. Regarding reviewer 3's point on similarity between GDF11-promoted and age-induced atrophy, although I will not be requiring you to perform additional experimentation on this specific point (provided all other issues are carefully and fully dealt with), I

do, however encourage you to develop it as far as realistically possible, and at the very least provide a detailed discussion.

Please note that it is EMBO Molecular Medicine policy to allow a single round of revision only and that, therefore, acceptance or rejection of the manuscript will depend on the completeness of your responses included in the next, final version of the manuscript.

As you know, EMBO Molecular Medicine has a "scooping protection" policy, whereby similar findings that are published by others during review or revision are not a criterion for rejection. However, I do ask you to get in touch with us after three months if you have not completed your revision, to update us on the status. Please also contact us as soon as possible if similar work is published elsewhere.

Please note that EMBO Molecular Medicine now requires a complete author checklist (<http://embomolmed.embopress.org/authorguide#editorial3>) to be submitted with all revised manuscripts. Provision of the author checklist is mandatory at revision stage; The checklist is designed to enhance and standardize reporting of key information in research papers and to support reanalysis and repetition of experiments by the community. The list covers key information for figure panels and captions and focuses on statistics, the reporting of reagents, animal models and human subject-derived data, as well as guidance to optimise data accessibility. The Author checklist will be published alongside the paper, in case of acceptance, within the transparent review process file.

Finally, we now mandate that all corresponding authors list an ORCID digital identifier. You may do so through our web platform upon submission and the procedure takes <90 seconds to complete. We also encourage co-authors to supply an ORCID identifier, which will be linked to their name for unambiguous name identification.

I look forward to seeing a revised form of your manuscript as soon as possible.

***** Reviewer's comments *****

Referee #1 (Remarks):

The work of Lee Sweeney group on GDF11 is addressing an important and critical issue related to the potential beneficial effect of GDF11 as anti-ageing factor. The author used in the first step muscle cell culture to prove that GDF11 and myostatin show similar biological effects in terms of muscle loss and that both activate the canonical Smad2/3 signalling. Then they moved in vivo and confirmed the pro-cachectic action of GDF11. The experimental design is very elegant, appropriate to address the important scientific issues about the GDF11 physiological relevance and conclusions are correctly interpreted. These findings are extremely relevant for therapeutic purpose. Authors may consider to address a minor point.

The authors correctly analysed the pathways that are linked to TGF β signalling. However to better complete the signalling picture, it would be of interest to know whether GDF11-mediated muscle wasting requires the activation of an atrophy program. A nice addition would be to monitor the expression levels of atrogenes such as MUSA1, Atrogin1, MuRF1, Ubiquitin, LC3 in heart and skeletal muscle of GDF11 treated mice.

Authors used the term frailty to describe the deleterious effect of GDF11 on muscle and heart mass. However the term frailty describes a human syndrome characterised by susceptibility to adverse events in elderly people. The phenotype of the treated mice much better fit cachexia (body weight loss consequent to lean and WAT wasting that is independent of nutritional status). Therefore, this reviewer suggests to change in the text the inappropriate word frailty with the more appropriate word cachexia.

Referee #2 (Remarks):

The authors have provided a comprehensive analysis of the effects of ectopically expressed GDF-11 on skeletal muscle and analyzed the signaling pathways involved. The data are a significant addition to the field and provide extensive support for a negative affect of GDF-11 on skeletal muscle growth. The study is comprehensive and an important contribution that has not yet been performed by other research groups. The authors provide extensive data for the effects of ectopically expressed GDF-11 on both cardiac and skeletal muscle.

I have few comments for the authors. Two experimental additions that could benefit the study. First, in Fig 2, is the reduction in myotube size accompanied by a reduction in myonuclear number? Is cell death occurring? Second is to determine whether the resultant atrophy from GDF-11 over-expression in skeletal muscle causes a reduction in the numbers of myofibers and whether a change in the numbers of myonuclei (myonuclear domain).

A few typos: line 96 though they remain? line 352: only began to exhibit? In Fig. 5B some of the p-p38 (the first p is cut off).

In summary the authors have presented a thorough and comprehensive manuscript that will be of high value and significance to the fields of aging, muscle atrophy and wasting as well as those working in the physiological roles of myostatin and GDF-11 in striated muscle.

Referee #3 (Comments on Novelty/Model System):

The authors make a conclusion that GDF11 promote muscle atrophy and frailty similar to aging condition. However, both their in vitro and in vivo fail to assess the aging part.

Referee #3 (Remarks):

The objective of this study is address the controversy regrading the role of GDF11 in age-mediated muscle atrophy. Indeed, recent papers have reported that GDF11 could be used as a therapeutic to stop or reverse age-induced muscle atrophy. However, other reports have reached the opposite conclusion. To address this, the authors used in vitro and in vivo models to assess the impact of GDF11 on both skeletal muscle and cardiac muscle. They also compared the effect of GDF11 to Mstn, another member of the transforming β superfamily, which has great homology with GDF11. In vitro both GDF11 and Mstn activate the SMAD2/3 pathway in HEK293 or C2C12 myoblasts and myotubes, 30 to 60 minutes upon exposer of. However, it takes 3 days for either of these factors to promote the atrophy of myotubes in vitro. Both Mstn and GDF11 activate the SMAD3 pathway. In vivo both GDF11 and Mstn promoted the atrophy of skeletal muscle, however only Gdf11 triggered the atrophy in cardiac muscle. Correlative observations linked these effects to a differential activation of downstream pathways and availability of Activin receptor.

The topic of the paper is very important and timely. Indeed, age-induced muscle loss is a huge health problem against which there is no available treatment. Providing definitive proof on the role of important factors such as GDF11 in muscle integrity and fate is crucial and will definitely help in designing novel therapeutic strategies. While the authors provide nice correlative data linking GDF11 and Mstn activity to muscle atrophy, the connection to the downstream effectors such as SMAD2/3 and ALK4/5 pathways is not convincing. Indeed, the in vitro data show that the GDF11- and Mstn-mediated activation of the SMAD2/3 pathway occurs as early as 30 min upon exposer. However, muscle atrophy requires 3 days of exposer to these factors. It is important to assess whether during the 3 days of exposer the SMAD2/3 pathway remain active. A more detailed study addressing this is required. The other issue is the lack of connection to age-induced atrophy. The author should explain how their data could be relevant to age-induced muscle atrophy. Another important aspect of the study is the differential effect of Mstn and GDF11 on cardiac muscle. In the discussion the authors indicate that this differential effect could be explained by the levels of Activin Receptor in skeletal muscle compared to cardiac muscle. This is an important conclusion that needs to be supported by more in vitro experiments.

Other important comments:

1) Figure 1: Using AlphaLISA Assays and recombinant Mstn and GDF11 the authors assessed the activation of the SMAD by the TGF λ 3 superfamily ligands that include Mstn and GDF11 on both HEK293 and C2C12 cells. They observed that in HEK293 cells Mstn and GDF11, similarly to TGF β , activate the SMAD2/3 but not the SMAD1/5/8 pathway through the ActIIBR receptor. In C2c12 myoblasts and myotubes however, both Mstn and GDF11 activate the SMAD2/3 pathway in a similar way although less potent than TGF β .

While the AlphaLISA data support the conclusion made regarding SMAD2/3 pathway activation by Mstn and GDF11 in both myoblasts and myotubes, the western blot results cast a doubt on this conclusion. Indeed, it is hard to be convinced that the observed bands in the top panel of figure 1C are indeed p-SMAD3. They should provide more convincing western blot to confirm activation of SMAD3. Additionally, it is critical to link SMAD2/3 activation observed in this experiments (30 to 60 min exposure to Mstn and GDF11) to the following experiments showed in Figure 2 and beyond, where the exposure of muscle fibers is much longer?

2) Figure 2: They exposed C2C12 muscle fibers to recombinant GDF11, Mstn and TGF β for 3 days and assessed muscle atrophy by measuring fiber diameter. The authors concluded that GDF11 promotes atrophy similarly to Mstn and TGF β . The immunofluorescence images not only show reduction in diameter after exposure to GDF11 but also indicate that these cells have much less fibers compared to the control and the other treatment. Why is this? Is this field representative of all experiments with GDF11? The authors stated that GDF11 prevent muscle differentiation ("As it has been previously demonstrated that recombinant GDF11 can inhibit myoblast differentiation in vitro 14, 17). Does GDF11-induced atrophy observed in this experiments uses the same downstream pathways than GDF11-mediated inhibition of muscle differentiation?

What is the status of the SMAD3 pathway in the wasted fibers seen in Figure 2? The authors showed in the previous figure that the activation of SMAD3 occurs as early as 30 min after exposure to GDF11. In figure 2 however, the muscle fibers where exposed for 3 days. It is important to show a direct correlation between SMAD3 activation and the observed wasting. What is the impact of 3 days treatment with GDF11 or Mstn on SMAD3 activation? It is crucial that the authors address this issue, since it is possible that Mstn- and GDF11-mediated phosphorylation of SMAD2/3 decreases with time of exposure, in which case this could indicate a SMAD2/3-independent effect.

3) Figures 3-4 Using liver specific α 1-anti-trypsin promoter packaged into AAV2/8 the authors overexpressed GDF11 and Mstn in mice and assessed muscle heart status. They observed that GDF11 causes atrophy in both skeletal and cardiac muscle, while Mstn promoted the atrophy only of skeletal muscle atrophy but not in the heart.

4) Figure 5: The authors show a differential activation of various pathways by GDF11 in skeletal muscle and the heart. However, the connection between the non-canonical pathways and the SMAD2/3 in mediated the observed atrophy is not explored. The conclusion the "This suggests that in cardiac muscle, the magnitude of SMAD3 signaling can be regulated by decreasing total SMAD3" is premature and based only on correlative data. A more in depth studies of the relationship between these pathways and GDF11-mediated atrophy is needed before making any conclusion. This could be easily addressed using an in vitro cell system.

5) Figure 6: Clearly showed that GDF11 and Mstn trigger similar muscle loss only GDF11 affects the heart. The data are clear and support the conclusion.

6) Figure 7: The main conclusion of this figure is that "At these expression levels, it is possible that GDF11 binding to ActIIBR may preferentially recruit ALK5 more so than Mstn, explaining the differential effects of the two ligands in the heart". In my opinion the data do not support such a conclusion. The authors followed the expression of the mRNA encoding for ALK4 and 5 not the proteins. An effect on mRNA expression does not always indicate and/or follow protein expression. Therefore, the connection between GDF11 and Mstn effects and these downstream effectors needs be demonstrated using in vitro system that mimic muscle atrophy as described above.

Overall, the topic of the study is important and the experiments are in general well done. However,

the authors need to provide mechanistic data as described above to support the conclusion that GDF11 promote muscle atrophy similarly to Mstn via the SMAD2/3 pathway and the activating receptor.

1st Revision - authors' response

05 January 2017

We very much appreciate the time and effort involved in the review of our manuscript and your comments. We feel that by responding to them, the revised version of the current manuscript is greatly improved.

Referee #1 (Remarks):

The work of Lee Sweeney group on GDF11 is addressing an important and critical issue related to the potential beneficial effect of GDF11 as anti-ageing factor. The author used in the first step muscle cell culture to prove that GDF11 and myostatin show similar biological effects in terms of muscle loss and that both activate the canonical Smad2/3 signalling. Then they moved in vivo and confirmed the pro-cachectic action of GDF11. The experimental design is very elegant, appropriate to address the important scientific issues about the GDF11 physiological relevance and conclusions are correctly interpreted. These findings are extremely relevant for therapeutic purpose. Authors may consider to address a minor point.

The authors correctly analysed the pathways that are linked to TGF β signalling. However to better complete the signalling picture, it would be of interest to know whether GDF11-mediated muscle wasting requires the activation of an atrophy program. A nice addition would be to monitor the expression levels of atrogenes such as MUSA1, Atrogin1, MuRF1, Ubiquitin, LC3 in heart and skeletal muscle of GDF11 treated mice.

Thank you for this excellent suggestion. In the current version of the manuscript (found in Figure 5D), we have included data on gene expression of the muscle-specific E3 ubiquitin ligases atrogin-1, MuRF1, and MUSA1 from quadriceps and hearts of 3 and 5-day high-dose AAV8.GDF11 exposure. While we do find significant elevations in atrogin-1 and MuRF1 in the quadriceps and MuRF1 in the heart, these increases are quite modest compared to those reported for other atrophy models (particularly those reported by Satchek et al. 2007)

Authors used the term frailty to describe the deleterious effect of GDF11 on muscle and heart mass. However the term frailty describes a human syndrome characterised by susceptibility to adverse events in elderly people. The phenotype of the treated mice much better fit cachexia (body weight loss consequent to lean and WAT wasting that is independent of nutritional status). Therefore, this reviewer suggests to change in the text the inappropriate word frailty with the more appropriate word cachexia.

We apologize for this misuse of “frailty”, and thank the reviewer for pointing this out. We intended the term to describe severe muscle wasting leading to impaired mobility, independent of age. As you suggest, we have replaced “frailty” with “cachexia” to avoid confusion with aging-related sarcopenia.

Referee #2 (Remarks):

The authors have provided a comprehensive analysis of the effects of ectopically expressed GDF-11 on skeletal muscle and analyzed the signaling pathways involved. The data are a significant addition to the field and provide extensive support for a negative affect of GDF-11 on skeletal muscle growth. The study is comprehensive and an important contribution that has not yet been performed by other research groups. The authors provide extensive data for the effects of ectopically expressed GDF-11 on both cardiac and skeletal muscle.

I have few comments for the authors. Two experimental additions that could benefit the study. First, in Fig 2, is the reduction in myotube size accompanied by a reduction in myonuclear number? Is cell death occurring? Second is to determine whether the resultant atrophy from GDF-11 over-

expression in skeletal muscle causes a reduction in the numbers of myofibers and whether a change in the numbers of myonuclei (myonuclear domain).

Thank you for these recommendations. While our research group is very interested in myonuclear dynamics, especially concerning myonuclear death, we have not observed any indication that GDF11 or myostatin substantially affect myonuclear number in vitro or in vivo. To demonstrate this in the current report, we have added myonuclear content data (found in Figure 2E), where the number of nuclei per μm of myotube length (since diameter is affected) is actually increased with myostatin and GDF11 treatments. From our observations, we suspect myonuclear content is maintained during ActRIIB-dependent atrophy in a similar manner to that described by Bruusgaard and Gundersen (2008), where myonuclei are maintained during rapid atrophy conditions. A dedicated study using the methods previously used by our group in collaboration with the Larsson laboratory (Qaisar et al. 2012) to quantify myonuclear domain would be best to address that specific question, which is outside the scope of the current work.

A few typos: line 96 though they remain? line 352: only began to exhibit? In Fig. 5B some of the p-p38 (the first p is cut off).

Thank you for noting these errors. They have been corrected.

In summary the authors have presented a thorough and comprehensive manuscript that will be of high value and significance to the fields of aging, muscle atrophy and wasting as well as those working in the physiological roles of myostatin and GDF-11 in striated muscle.

Referee #3 (Comments on Novelty/Model System):

The authors make a conclusion that GDF11 promote muscle atrophy and frailty similar to aging condition. However, both their in vitro and in vivo fail to assess the aging part.

Once again, we apologize for the misuse of the term “frailty” to describe the severe wasting phenotype observed in our experiments. As suggested by Reviewer #1, we have replaced “frailty” with “cachexia” to avoid confusion with aging-related decrements. This study was never meant to address aging per se. Aging was mentioned because of the previously proposed approach of using GDF11 as an “anti-aging” therapeutic. This study demonstrates that contrary to the initial report, GDF11 drives atrophy in skeletal muscle as well as in the heart. Thus using GDF11 to counter pathological cardiac hypertrophy will drive skeletal muscle atrophy. This is obviously not advisable in an elderly population where maintenance of skeletal muscle mass and strength is a major health problem. We are not claiming that GDF11 contributes to age-related muscle atrophy. We are demonstrating that the use of exogenous GDF11 will drive muscle atrophy.

Referee #3 (Remarks):

The objective of this study is address the controversy regrading the role of GDF11 in age-mediated muscle atrophy. Indeed, recent papers have reported that GDF11 could be used as a therapeutic to stop or reverse age-induced muscle atrophy. However, other reports have reached the opposite conclusion. To address this, the authors used in vitro and in vivo models to assess the impact of GDF11 on both skeletal muscle and cardiac muscle. They also compared the effect of GDF11 to Mstn, another member of the transforming β superfamily, which has great homology with GDF11. In vitro both GDF11 and Mstn activate the SMAD2/3 pathway in HEK293 or C2C12 myoblasts and myotubes, 30 to 60 minutes upon exposure. However, it takes 3 days for either of these factors to promote the atrophy of myotubes in vitro. Both Mstn and GDF11 activate the SMAD3 pathway. In vivo both GDF11 and Mstn promoted the atrophy of skeletal muscle, however only GDF11 triggered the atrophy in cardiac muscle. Correlative observations linked these effects to a differential activation of downstream pathways and availability of Activin receptor.

The topic of the paper is very important and timely. Indeed, age-induced muscle loss is a huge health problem against which there is no available treatment. Providing definitive proof on the role of important factors such as GDF11 in muscle integrity and fate is crucial and will definitively help

in designing novel therapeutic strategies. While the authors provide nice correlative data linking GDF11 and Mstn activity to muscle atrophy, the connection to the downstream effectors such as SMAD2/3 and ALK4/5 pathways is not convincing. Indeed, the in vitro data show that the GDF11- and Mstn-mediated activation of the SMAD2/3 pathway occurs as early as 30 min upon exposure. However, muscle atrophy requires 3 days of exposure to these factors. It is important to assess whether during the 3 days of exposure the SMAD2/3 pathway remains active. A more detailed study addressing this is required.

Thank you for expressing this concern, as we should have pointed out that in the steady state, the signaling via the SMAD2/3 pathway will be blunted, but the expectation is that it will be much more elevated acutely when GDF11 (or Mstn) are initially applied. We now better show the elevation of Smad2/3 phosphorylation at 30 and 60 minutes (Fig. 1C), which bears out this point. The requirement of the SMAD2/3 pathway in ActRIIB-dependent muscle atrophy (both in vitro and in vivo) has been previously demonstrated (Sartori et al. 2009 and Trendelenburg et al. 2009). Three days of exposure to ligands is not a requirement for atrophy demonstration by the administered ligands, but rather is our standardized protocol to evaluate steady-state phenotypes of administered molecules on myotube size, in vitro. This has been more clearly worded in the current version of the manuscript. Additionally, we have included p-SMAD3 blots from these steady state cultures to demonstrate that the pathway is still elevated (Figure 2F), albeit not to the degree of acute administration (which is typical of steady-state cultures).

The other issue is the lack of connection to age-induced atrophy. The author should explain how their data could be relevant to age-induced muscle atrophy.

As mentioned above, we did not intend to compare the phenotype observed in our studies to age-related atrophy, therefore have removed all mention of “frailty” from the phenotype description to avoid this confusion. We are not arguing that GDF11 causes age-related atrophy. We are arguing that it can never be considered as an “anti-aging” therapeutic because it will cause striated muscle atrophy.

Another important aspect of the study is the differential effect of Mstn and GDF11 on cardiac muscle. In the discussion the authors indicate that this differential effect could be explained by the levels of Activin Receptor in skeletal muscle compared to cardiac muscle. This is an important conclusion that needs to be supported by more in vitro experiments.

For clarification, we are offering the hypothesis that differential preference of these ligands for ALK4 and ALK5 may be involved in the differential effects they have on cardiac muscle, where high levels of GDF11 may end up outcompeting TGFβ1 for ALK5 occupancy on cardiomyocytes and resulting in cardiac atrophy. We have now included data that further build this hypothesis, as GDF11-exposed hearts strongly upregulate Tgfb1 (Figure 7G), suggesting possible compensation for interference by GDF11. We agree that more experiments are required to fully test this hypothesis, which are better suited as a separate investigation.

Other important comments:

1) Figure 1: Using AlphaLISA Assays and recombinant Mstn and GDF11 the authors assessed the activation of the SMAD by the TGF λ 3 superfamily ligands that include Mstn and GDF11 on both HEK293 and C2C12 cells. They observed that in HEK293 cells Mstn and GDF11, similarly to TGF β , activate the SMAD2/3 but not the SMAD1/5/8 pathway through the ActRIIB receptor. In C2c12 myoblasts and myotubes however, both Mstn and GDF11 activate the SMAD2/3 pathway in a similar way although less potent than TGF β . While the AlphaLISA data support the conclusion made regarding SMAD2/3 pathway activation by Mstn and GDF11 in both myoblasts and myotubes, the western blot results cast a doubt on this conclusion. Indeed, it is hard to be convinced that the observed bands in the top panel of figure 1C are indeed p-SMAD3. They should provide more convincing western blot to confirm activation of SMAD3. Additionally, it is critical to link SMAD2/3 activation observed in this experiments (30 to 60 min exposure to Mstn and GDF11) to the following experiments showed in Figure 2 and beyond, where the exposure of muscle fibers is much longer?

Thank you for voicing this concern. Better quality immunoblots for SMAD2/3 activation have been included (Figure 1C). SMAD3 activation in the steady-state cultures has also been displayed (Figure 2F).

2) Figure 2: They exposed C2C12 muscle fibers to recombinant GDF11, Mstn and TGF β for 3 days and assessed muscle atrophy by measuring fiber diameter. The authors concluded that GDF11 promotes atrophy similarly to Mstn and TGF β . The immunofluorescence images not only show reduction in diameter after exposure to GDF11 but also indicate that these cells have much less fibers compared to the control and the other treatment. Why is this? Is this field representative of all experiments with GDF11? The authors stated that GDF11 prevent muscle differentiation ("As it has been previously demonstrated that recombinant GDF11 can inhibit myoblast differentiation in vitro 14, 17). Does GDF11-induced atrophy observed in this experiments uses the same downstream pathways than GDF11-mediated inhibition of muscle differentiation? What is the status of the SMAD3 pathway in the wasted fibers seen in Figure 2? The authors showed in the previous figure that the activation of SMAD3 occurs as early as 30 min after exposure to GDF11. In figure 2 however, the muscle fibers were exposed for 3 days. It is important to show a direct correlation between SMAD3 activation and the observed wasting. What is the impact of 3 days treatment with GDF11 or Mstn on SMAD3 activation? It is crucial that the authors address this issue, since it is possible that Mstn- and GDF11-mediated phosphorylation of SMAD2/3 decreases with time of exposure, in which case this could indicate a SMAD2/3-independent effect.

For these assays, myoblasts were differentiated for 7 days prior to the administration of any exogenous ligand for the purpose avoiding any effects on myoblast differentiation (which do involve SMAD2/3 signaling, as described by Trendelenburg et al. 2009 and Egerman et al. 2014). The representative images in Figure 2B were selected based on clear demonstration of mean fiber diameter rather than number of fibers per field of view. Additionally (as mentioned in response to Reviewer #2), we do not see any evidence that myonuclear number is affected by administration of myostatin or GDF11. As mentioned above, SMAD3 activation from this experiment is now demonstrated in Figure 2F.

3) Figures 3-4 Using liver specific α 1-anti-trypsin promoter packaged into AAV2/8 the authors overexpressed GDF11 and Mstn in mice and assessed muscle heart status. They observed that GDF11 causes atrophy in both skeletal and cardiac muscle, while Mstn promoted the atrophy only of skeletal muscle atrophy but not in the heart.

4) Figure 5: The authors show a differential activation of various pathways by GDF11 in skeletal muscle and the heart. However, the connection between the non-canonical pathways and the SMAD2/3 in mediated the observed atrophy is not explored. The conclusion the "This suggests that in cardiac muscle, the magnitude of SMAD3 signaling can be regulated by decreasing total SMAD3" is premature and based only on correlative data. A more in depth studies of the relationship between these pathways and GDF11-mediated atrophy is needed before making any conclusion. This could be easily addressed using an in vitro cell system.

Based on your concerns and in light of new data depicting cardiac p-SMAD3 activation at earlier time points in vivo (Figure 5C), this speculative statement/line of thought has been modified for better clarity.

5) Figure 6: Clearly showed that GDF11 and Mstn trigger similar muscle loss only GDF11 affects the heart. The data are clear and support the conclusion.

6) Figure 7: The main conclusion of this figure is that "At these expression levels, it is possible that GDF11 binding to ActIIBR may preferentially recruit ALK5 more so than Mstn, explaining the differential effects of the two ligands in the heart". In my opinion the data do not support such a conclusion. The authors followed the expression of the mRNA encoding for ALK4 and 5 not the proteins. An effect on mRNA expression does not always indicate and/or follow protein expression. Therefore, the connection between GDF11 and Mstn effects and these downstream effectors needs be demonstrated using in vitro system that mimic muscle atrophy as described above.

While it is true that gene expression does not always reflect protein content, it does provide valuable information on cellular responses, especially considering these genes are likely post-

transcriptionally regulated similarly in skeletal and cardiac muscle. Unfortunately, we have not yet identified suitable anti-ALK4 or anti-ALK5 antibodies that meet our validation standards for immunoblotting to verify protein content in tissue lysates.

Overall, the topic of the study is important and the experiments are in general well done. However, the authors need to provide mechanistic data as described above to support the conclusion that GDF11 promote muscle atrophy similarly to Mstn via the SMAD2/3 pathway and the activating receptor.

We feel that the combination of the in vivo data and in vitro data presented now clearly show that both Mstn and GDF-11 signal through the SMAD2/3 pathway in the heart and in skeletal muscle.

2nd Editorial Decision

25 January 2017

Thank you for the submission of your revised manuscript to EMBO Molecular Medicine. We have now received the enclosed report from the reviewer who was asked to re-assess it. As you will see s/he is now globally supportive and I am pleased to inform you that we will be able to accept your manuscript pending the following final amendments:

- 1) While performing our pre-publishing quality control and image screening routines, we noticed that several main figure panels are duplicated in the EV figures. Please explain these occurrences. We understand that in most cases the EV figures might be presenting full corresponding datasets, but please clearly explain such occurrences in the figure legends to avoid any misunderstandings.
- 2) The manuscript must include a statement in the Materials and Methods identifying the institutional and/or licensing committee approving the experiments, including any relevant details (like how many animals were used, of which gender, at what age, which strains, if genetically modified, on which background, housing details, etc). We encourage authors to follow the ARRIVE guidelines for reporting studies involving animals. Please see the EQUATOR website for details: <http://www.equator-network.org/reporting-guidelines/improving-bioscience-research-reporting-the-arrive-guidelines-for-reporting-animal-research/>. Please make sure that ALL the above details are reported and amend the checklist as appropriate.
- 3) Data described in submitted manuscripts should be deposited in a MIAME-compliant format with one of the public databases. We would therefore ask you to submit your microarray data to the ArrayExpress database maintained by the European Bioinformatics Institute for example. ArrayExpress allows authors to submit their data to a confidential section of the database, where they can be put on hold until the time of publication of the corresponding manuscript. Please see <http://www.ebi.ac.uk/arrayexpress/Submissions/> or contact the support team at arrayexpress@ebi.ac.uk for further information.
- 4) We encourage the publication of source data, with the aim of making primary data more accessible and transparent to the reader. Would you be willing to provide a PDF file per figure that contains the original, uncropped and unprocessed scans of all or at least the key gels used in the manuscript and/or source data sets for relevant graphs? The files should be labeled with the appropriate figure/panel number, and in the case of gels, should have molecular weight markers; further annotation may be useful but is not essential. The files will be published online with the article as supplementary "Source Data" files. If you have any questions regarding this just contact me.

Please submit your revised manuscript within two weeks. I look forward to seeing a revised form of your manuscript as soon as possible.

***** Reviewer's comments *****

Referee #3 (Comments on Novelty/Model System):

The quality of the data as well as the model used are adequate and the revised version addressed all

the technical issues I raised before.

The medical relevance of the study since addressing GDF11 role in muscle formation and atrophy could have consequence on treating disease induced muscle loss.

Referee #3 (Remarks):

I thank the authors for their efforts in addressing my comments. I am now satisfied with the revised version

2nd Revision - authors' response

26 January 2017

As you will find, the current version of the manuscript addresses the editorial comments detailed in your decision letter. Specifically, it is clarified in the Expanded View Figure captions that Fig EV2 and EV4 contain the source data for Fig 3 and Fig 5, respectively. Also, we have included a Synopsis section, added molecular weight information to Fig EV4, and expanded our animal use methods to accommodate ARRIVE guidelines.

Corresponding Author Name: H. Lee Sweeney

Manuscript Number: EMM-2016-07231